# Chronological Analysis of Primary Cervical Spine Infection: A Single-Center Analysis of 59 Patients over Three Decades (1992–2018)

**DOI:** 10.3390/jcm11082210

**Published:** 2022-04-15

**Authors:** Myung-Jin Sung, Sung-Kyu Kim, Hyoung-Yeon Seo

**Affiliations:** 1Department of Orthopaedic Surgery, Chonnam National University Hospital, Gwangju 61469, Korea; smj2384@naver.com (M.-J.S.); 1976skkim@daum.net (H.-Y.S.); 2Department of Orthopaedic Surgery, Chonnam National University Medical School, Gwangju 61469, Korea

**Keywords:** cervical spine, primary infection, chronological, outcome

## Abstract

Primary cervical spine infections progress quickly and cause neurological impairment at an early stage. Despite their clinical significance, few studies have investigated primary cervical spine infections, owing to the rarity of the condition. This study analyzed the characteristics of 59 patients treated for primary cervical spine infections between 1992 and 2018 at our hospital. Clinical and radiological analyses were conducted. Moreover, a comparative analysis was performed, incorporating each patient’s underlying disease, mortality and complications, and treatment results. Comparison between groups based on the chronological period (1992–2000, 2001–2009, and 2010–2018) revealed that the mean age of onset has increased significantly in recent years. The rate of neurological impairment, duration of antibiotic use, and frequency of underlying disease increased significantly with time. No significant differences among groups were observed in the hematological and microbiological analyses. The incidence rate of epidural abscess and multisegmental infection increased significantly in recent years. There was no statistically significant difference in the complication and mortality rates, according to the time period. We think that prompt diagnosis and appropriate treatment are necessary, considering the current trends in primary cervical spine infection.

## 1. Introduction

Primary spinal infections are uncommon, accounting for only 1–4% of all patients with primary osteomyelitis [1,2]. Primary cervical spine infections comprise only 6% of all spinal infections and are rarer than thoracic and lumbar segment infections [1,3,4,5,6]. However, cervical spine infections are known to increase morbidity and mortality rates since they cause neurological impairment at an early stage owing to their faster progression compared to other segmental spinal infections [1,5,7]. The incidence rate of primary cervical spine infections has risen recently due to the aging of the population and the increase in the incidence of chronic diseases that reduce immunity [8,9].Few studies have investigated primary cervical spine infections due to the scarcity of patients, despite their clinical significance. Therefore, this study aimed to analyze the characteristics of patients with primary cervical spine infections over the past three decades, followed by a comparison for each time period.

## 2. Materials and Methods

The medical records of patients who visited our hospital between 1992 and 2018 were reviewed retrospectively. The International Classification of Diseases, Ninth Revision, Clinical Modification codes and diagnoses retrieved from the hospital electronic medical records included “cervical spine infection”, “cervical spine spondylitis”, “cervical spine spondylodiscitis”, “cervical spine discitis”, and “cervical spine epidural abscess”. Patients with a secondary spinal infection due to trauma, surgery, or tumor invasion and patients whose hospital medical records lacked radiological data were excluded from this study. Finally, 59 patients with primary cervical spine infections were included. The patients were divided into three groups based on 9-year intervals: 1992 to 2000 (group A), 2001 to 2009 (group B), and 2010 to 2018 (group C) (Figure 1). Blood cultures and, if possible, bone biopsies were performed in all patients. Clinical analysis was conducted with the American Spinal Cord Injury Association (ASIA) impairment scale at the first and last follow-up visits to observe the changes in the neurological symptoms. The main symptoms and presence of concomitant non-spinal infection, antibiotic treatment duration, underlying disease, mortality and complications of each patient, and treatment method were reviewed. Hematological and microbiological analyses entailed the examination of the erythrocyte sedimentation rate (ESR), C-reactive protein (CRP), differential white blood cell (WBC) count, and the presence of pathogens. Plain anteroposterior, lateral, and flexion–extension radiographs of the cervical spine were acquired for the radiological analysis. The location of an infection relative to the spinal cord (anterior: anterior epidural space, vertebral body, or intervertebral disc; posterior: posterior epidural space, lamina, spinous process, or transverse process), presence of epidural abscesses, level of segmental involvement, and multisegmental infections were identified via contrast-enhanced magnetic resonance imaging (MRI) or computed tomography (CT).

Data were analyzed using the SPSS version 18.0 for Windows software (IBM Corporation, Armonk, NY, USA). Continuous data were expressed as means and standard deviations, and categorical variables were expressed as percentages. The Kruskal–Wallis test was performed to analyze the continuous variables, and the linear-by-linear association test was used to analyze the categorical variables. *p*-values less than 0.05 were considered statistically significant.

## 3. Results

### 3.1. Demographic Analysis

The study population included 42 male patients (71%) and 17 female patients (29%). The mean duration of outpatient follow-up was 35.8 months (14–267.2), and the mean age at diagnosis was 61.4 years. The chronological classification of the study population was as follows: groups A (1992–2000), B (2001–2009), and C (2010–2018) containing 9 (15.3%), 13 (22.0%), and 37 patients (62.7%), respectively. The number of patients diagnosed with a primary cervical spine infection was found to have increased in recent years. The mean age of the patients in groups A, B, and C was 54.1, 58.6, and 64.2 years, respectively, and increased significantly with time (*p* = 0.016). The proportion of men was predominant in all groups (*p* = 0.918) (Table 1).

### 3.2. Clinical Analysis

Posterior neck pain was the most common clinical symptom in all groups (total: 83.1%, group A: 77.8%, group B: 92.3%, group C: 81.1%), followed by fever (≥37.5°), chills, fatigue, and dysphagia (Table 2). The rate of neurological impairment at the time of admission (ASIA scale grades A–D) was 22.2% in group A, 30.8% in group B, and 62.2% in group C. The rate of neurological impairment was statistically significantly higher in group C (*p* = 0.012). Two patients with neurological impairment in group A recovered from the ASIA grade D to grade E. Two patients from group B recovered from the ASIA grade B to D and grade C to D, respectively. In group C, one patient recovered from the ASIA grade B to D, three patients recovered from the ASIA grade C to D, and three patients recovered from the ASIA grade D to E, but one patient’s condition deteriorated from the ASIA grade E to D (Table 3). Moreover, several patients had underlying diseases and risk factors that affected other organs and the immune system, and the proportion of such patients increased (statistically) significantly in recent years (group A: 11.1%, group B: 38.5%, group C: 64.9%, *p* = 0.002) (Table 1 and Table 4). None of the patients in group A and only one patient (pneumonia) in group B had a concomitant non-spinal infection. However, nine patients (24.3%) in group C had a concomitant non-spinal infection, which was statistically significant (urinary tract infection: three patients, infectious endocarditis: two patients, pneumonia: one patient, aortitis: one patient, epididymitis: one patient, and polyarticular septic arthritis: one patient) (*p* = 0.049). The duration of antibiotic treatment also rose statistically significantly in recent years (*p* = 0.033). However, one patient each from groups B and C, who had been treated for tuberculosis for more than 9 months, was excluded from the statistical analysis. The above-mentioned clinical analyses are summarized in Table 1.

### 3.3. Hematological and Microbiological Analyses

There was no difference in the hematological analysis with parameters such as ESR, CRP, and WBC among the three groups (Table 1). None of the parameters included in the microbiological analysis was statistically significant. Pathogens were isolated in five patients (38.5%) in group B and eight patients (21.6%) in group C, but not in group A. The most common pathogen was methicillin-sensitive *Staphylococcus aureus*, which accounted for 60% of the isolated pathogens in group B and 50% in group C. *Mycobacterium tuberculosis* was also identified in one patient each in group B and group C. The causative pathogen could not be isolated in 46 patients (78%) during the entire study period (Table 1).

### 3.4. Radiological Analysis

The results of the radiological analysis are summarized in Table 5. The C5–C6 segment was the most commonly involved site in the entire study population (47.5%). It was also the most common site of occurrence in each group stratified according to the period. The C1-C2 segment was the least-affected site (1.7%). MRI revealed that the infection was located in the anterior spinal cord in all groups, except for three patients (two patients: posterior, one patient: anterior and posterior) (total: 94.9%, group A: 100%, group B: 92.3%, and group C: 94.6%). Epidural abscess formation was observed in 24 patients (40.7%) on their MRI (Figure 2). An epidural abscess was absent in group A but increased to 3 patients (23.1%) in group B and 21 patients (56.8%) in group C. Twenty patients (33.9%) had a multisegmental infection, and its frequency of distribution was as follows: 1 patient (11.1%) in group A, which increased to 2 patients (15.4%) in group B and 17 patients (45.9%) in group C. The incidence of epidural abscess formation and multisegmental infection rose significantly in recent years (epidural abscess: *p* = 0.001, multisegmental infection: *p* = 0.017).

### 3.5. Treatment Method Analysis

Antibiotics were administered orally or intravenously to all patients. First-generation cephalosporins were used as empirical therapy after consultation with the infectious disease department if the causative pathogen could not be differentiated by culture. After identification, the therapeutic agent was switched to a suitable antibiotic to which the pathogen was susceptible. The duration of antibiotic use was determined after consultation with an infectious disease specialist. Patients were treated with antibiotics while wearing a soft-collar neck brace during the treatment period. The indications for surgery included unbearable posterior neck pain, bacteremia, formation of a perivertebral or epidural abscess, severe vertebral body and endplate destruction with local kyphosis and instability, and neurological impairment. Appropriate antibiotic treatment was administered for 4–6 weeks after surgical treatment. Fourteen of fifty-nine patients underwent conservative treatment, and forty-five underwent surgical treatment. Only four patients who underwent surgical treatment died. All patients in groups A and B received surgical treatment. Fourteen patients (37.8%) received conservative treatment, and 23 (62.2%) received surgical treatment in group C. Treatment was successful in all 14 patients who underwent conservative treatment. Successful treatment was also achieved in 42 (93.3%) of the 45 patients who underwent surgical treatment.

### 3.6. Complications

Complications including death (4 patients) occurred in 12 patients. The causes of death in four patients from group C were as follows: two patients died within 1 year postoperatively due to advanced bacteremia (one patient at 1 month and one patient at 11 months), and two patients died approximately 2 years after surgery from diseases that were not related to the cervical spine infection. The other complications included bone graft fixation failure in two patients, screw loosening in two patients, reoperation due to treatment failure, superficial surgical site infection, esophageal fistula, dysphagia, and voice change in one patient. Only one patient in group C displayed neurological deterioration from ASIA grade E to D. Neurological impairment occurred in the patient with bone graft fixation failure, and esophageal fistula formation occurred in the patient with a superficial surgical site infection. No statistically significant difference was observed in the rate of complications, including death, among groups A, B, and C (Table 6).

## 4. Discussion

### 4.1. Incidence Rate

For a long time, primary infections of the thoracic and lumbar spine have been reported frequently, although uncommonly. However, since primary cervical spine infections are even rarer, few studies have investigated this condition, and the number of cases studied remains small [1,2,10,11,12,13,14,15,16,17]. For this rare type of cervical spine infection, we attempted to analyze the flow and results of the past 30 years at our institution and compare the results according to the chronological period. A total of 9 (group A), 13 (group B), and 37 patients (group C) were treated from 1992 to 2000, 2001 to 2009, and 2010 to 2018, respectively. Calculation of the incidence by division of each period into one-year intervals revealed that the number of patients was 1, 1.5, and 4.2 per year in groups A, B, and C, respectively, demonstrating an increasing trend in the incidence. These changes may be attributed to the aging of the population, which is supported by the following evidence: the mean age in group A was 54.1 years, whereas that in groups B and C increased to 58.6 years and 64.2 years, respectively. Shousha et al. conducted a literature review of studies on cervical spine infection and found that, when the average age was calculated by dividing the preceding literature before and after 2004, the mean age was below 60 years before 2004 but exceeded 60 years after 2004. They reported that aging greatly influences the rise in the frequency of primary cervical spine infections [18]. The rates of alcohol consumption, smoking, and underlying diseases such as hypertension and diabetes were the highest in group C in our study. This result can also be confirmed by other studies that reported that underlying diseases and conditions that impair the immune system increase the frequency of primary cervical spine infections [19,20]. In our study, 42 of the 59 patients with primary cervical spine infections were men, accounting for 71% of the study population. Other studies also reported that cervical spine infections were predominant in men [1,13,14,17], which is thought to result from chronic factors such as drinking, smoking, or substance abuse. A meta-analysis of 915 patients with spinal infection reported a male–to–female ratio of 1:0.56, demonstrating male predominance. It was hypothesized that this tendency might result from the higher frequency of exposure to risk factors that are more common in men, such as trauma, substance abuse, and alcohol. Moreover, one study reported that most alcoholics consume a protein-deficient diet, which compromises their immune system [21]. Although it could not be confirmed in our study that analyzed primary cervical spine infection, other studies identified that the increase of non-invasive procedures, such as nerve block using steroids and intra-discal injection, may also be factors influencing secondary cervical spine infections [5].

On the other hand, C5–C6 was the most-affected segment, which remained consistent over time. Similar results are also found in several other studies and are related to the fact that the C5–C6 segment in the cervical vertebrae exhibits the greatest mobility and is susceptible to degenerative changes and infection. [3,17,22,23,24,25]. In contrast, C1–C2 was the least-affected segment, which was involved in only one patient in group B. The rarity of atlantoaxial (C1–C2) infection has also been confirmed by several other studies [12,22,26] since it is more difficult to diagnose on plain radiography compared to infections of other cervical spinal segments. Moreover, the actual rate of diagnosis is known to be low because the symptoms are not clear and the pain is not severe, prolonging the time interval between infection and diagnosis [1,26].

In this study, the causative pathogens were isolated only in groups B (38.5%) and C (21.5%). We suppose that the pathogen was not differentiated in 78% of the overall patient population due to the administration of antibiotics at other hospitals. Clinical symptoms of cervical spine infections, such as neck pain, were ambiguous and difficult to distinguish from the neck pain of the degenerative cervical spine disease. Of course, neck pain caused by infection is characterized as constant and continuous, but hematological tests and imaging tests such as MRI are ultimately necessary. However, if there is constant neck pain that is different from that in degenerative disease alone or manifests together with fever and/or chill, it is necessary to consider cervical spine infections for further evaluation.

### 4.2. Complications and Mortality

Cervical spine infection is a serious disease with significant complications, despite the advances in pharmacotherapy and surgery, and this is thought to arise from the specificity of the cervical spine. The cervical spine has a greater range of motion, and the spinal cord occupies a large portion of the spinal canal in this region, compared to the thoracic and lumbar segments. These characteristics make cervical spine infection potentially dangerous, owing to the faster progression of infection, which is more likely to cause neurological symptoms than the involvement of other segments [1,22,23]. Our study found that cervical spine infections tend to be more aggressive at present compared to in the past, which is supported by numerous clinical and radiological findings. From clinical results, the rate of neurological impairment tended to increase from group A to groups B and C. In other non-vertebral infections, there were no cases of infection in other organs or septic shock in group A and only one case (7.1%) in group B. However, nine patients (24.3%) in group C had non-spinal infections. From the radiological results, the rate of multisegmental cervical spine infection increased gradually in recent years. An epidural abscess was not observed in group A; however, its occurrence rapidly increased to 23.1% in group B and 56.8% in group C. These results suggest that cervical spine infections are becoming more and more dangerous in recent years, which may be related to the increase in above-mentioned chronic diseases, aging of the population, and indiscriminate use of antibiotics. Patients with underlying chronic diseases, such as heart disease, liver disease, diabetes, and kidney failure, are being treated with ever-evolving medical technology. It is possible that these patients may allow the spread of bacteria through the bloodstream due to increasingly more prevalent new medical technologies such as nerve block, neuroplasty, and nucleoplasty [4,11,27,28,29,30]. Shousha et al. compared patients with cervical spine infections between 1994 and 1999 and between 2004 and 2009: the 2004–2009 group showed a considerably higher rate of underlying diseases, which could have contributed to the increased susceptibility to bacterial infection [18].

### 4.3. Limitations and Strengths

This study has a few limitations. First, the analysis conducted in this study was performed on patients from a single center. Therefore, the findings may not apply to all patients with primary cervical spine infections. These limitations could be overcome by conducting a multicenter study. Second, selection bias was inherent owing to the non-randomized and retrospective design of this study. Additionally, this study relied on the review of medical records.

Nevertheless, this study is meaningful since it evaluated the largest number of patients with primary cervical spine infections who were treated using the same method at a single center over a long period of approximately 30 years and stratified by a specific period. Although there was a difference in the number of patients between the groups, this is considered to be a tendency according to the timing of the primary cervical spine infections.

## 5. Conclusions

Although primary cervical spine infections are rare, their incidence rate has increased in the present day compared to the past. Epidural abscess formation, multisegmental involvement, complications, and mortality have also increased in recent years. Primary cervical spine infections have shown a more aggressive tendency in recent years. We suggest that rapid diagnosis and appropriate treatment are necessary, considering this tendency of primary cervical spine infections.

## Figures and Tables

**Figure 1 jcm-11-02210-f001:**
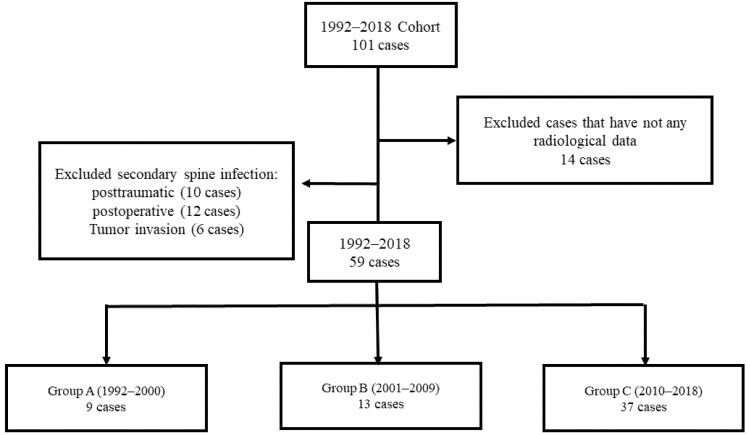
Patient selection and grouping flow.

**Figure 2 jcm-11-02210-f002:**
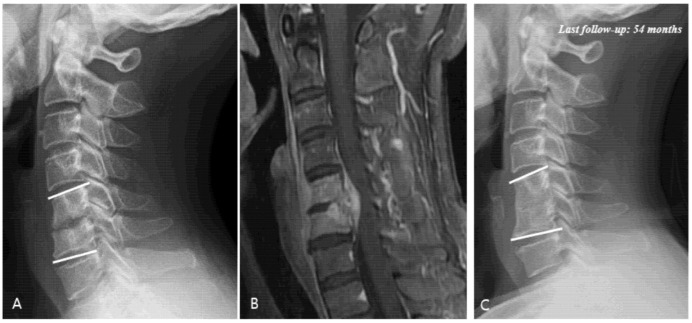
(**A**) Simple radiographs of 51-year-old male showing C5–C6 disc space narrowing and osteolytic change. (**B**) Gadolinium-enhanced magnetic resonance image showing infectious spondylodiscitis at C5–C6 accompanied by prevertebral abscess and anterior epidural abscess. (**C**) Last follow-up (54 months) plain radiographs showing fusion state after antibiotic treatment.

**Table 1 jcm-11-02210-t001:** Clinical, hematological and microbiological analysis.

Parameter	Total (*n* = 59)	Group A (*n* = 9)	Group B (*n* = 13)	Group C (*n* = 37)	*p*-Value
**Mean age (year)**	61.4 ± 10.0	54.1 ± 3.9	58.6 ± 8.6	64.2 ± 10.4	0.016 *
**Sex (male/female)**	42/17	6/3	9/4	27/10	0.683 †
**Antibiotic treatment duration (day)**	47.4 ± 15.3 (*n* = 57) **	38.9 ± 6.9	40.8 ± 11.0 (*n* = 11) **	51.67 ± 16.50 (*n* = 36) **	0.033 *
**Non-spinal infection (*n*)**	10	0	1	9	0.049 †
**Underlying disease & risk factor (*n*)**	30	1	5	24	0.002 †
**WBC (µL)**					
First visit	10,333.1 ± 4318.9	9883.3 ± 4193.2	9269.2 ± 2752.1	10,816.2 ± 4787.6	0.627 *
Last follow-up	7027.1 ± 3504.1	4522.2 ± 939.1	7376.9 ± 3314.4	7513.5 ± 3750.9	0.008 *
**CRP (mg/L)**					
First visit	8.8 ± 8.1	8.7 ± 5.3	11.1 ± 9.8	8.0 ± 8.0	0.562 *
Last follow-up	2.2 ± 4.4	0.8 ± 0.7	1.4 ± 1.7	2.8 ± 5.4	0.975 *
**ESR (mm/h)**					
First visit	80.9 ± 30.8	92.3 ± 23.2	89.2 ± 38.2	75.2 ± 28.9	0.159 *
Last follow-up	40.0 ± 30.1	31.3 ± 23.2	43.9 ± 32.7	40.7 ± 31.9	0.755 *
**Pathogen**	13/59	0/9	5/13	8/37	0.444 †
MRSA	1	0	1	0	
MSSA	7	0	3	4	
Gram negative	1	0	0	1	
Others	4	0	1	3	

Continuous variables are expressed as mean (SD). WBC = White blood cell, CRP = C-reactive protein, ESR = Erythrocyte sedimentation rate (ESR), MRSA = Methicillin-resistant *Staphylococcus aureus*, MSSA = Methicillin-susceptible *S. aureus*. * Kruskal–Wallis test, † Linear by linear association. ** excluded 1 patient each due to tuberculosis infection.

**Table 2 jcm-11-02210-t002:** Main symptoms analysis.

Main Symptom	Total (*n* = 59)	Group A (*n* = 9)	Group B (*n* = 13)	Group C (*n* = 37)	*p*-Value
Posterior neck pain	49 (83.1%)	7	12	30	0.906 †
Fever (≥37.5°)	42 (71.2%)	4	10	28	0.119 †
Chills	37 (62.7%)	5	6	26	0.217 †
Fatigue	15 (25.4%)	4	5	12	0.477 †
Dysphagia	3 (5.1%)	1	0	2	0.738 †

† Linear by linear association.

**Table 3 jcm-11-02210-t003:** Neurologic symptom analysis.

	Group A (*n* = 9)	Group B (*n* = 13)	Group C (*n* = 37)	*p*-Value
**Neurologic impairment**				
First visit	2	4	23	0.012 †
Last follow-up	0	4	21	0.001 †
**ASIA impairment scale**	First visit → Last follow-up			
Grade A	0 → 0	0 → 0	0 → 0	
Grade B	0 → 0	1 → 0	3 → 2	
Grade C	0 → 0	1 → 0	4 → 1	
Grade D	2 → 0	2 → 4	16 → 18	
Grade E	7 → 9	9 → 9	14 → 16	

† Linear by linear association.

**Table 4 jcm-11-02210-t004:** Presence of underlying diseases and risk factors.

Underlying Disease and Risk Factor	Group A (*n* = 9)	Group B (*n* = 13)	Group C (*n* = 37)	Total (*n* = 59)
Hypertension	1	4	19	24
Diabetes	0	4	9	13
Kidney failure	0	1	3	4
Heart disease	0	0	4	4
Liver disease	0	0	2	2
Cerebral infarction	0	0	2	2
Tumor	0	1	2	3
Alcohol	0	0	6	6
Smoking	1	2	9	12

**Table 5 jcm-11-02210-t005:** Radiological analysis.

Segment	Group A (*n* = 9)	Group B (*n* = 13)	Group C (*n* = 37)	Total (*n* = 59)
C1–C2	0	1	0	1 (1.7%)
C2–C3	1	0	3	4 (6.8%)
C3–C4	0	3	9	12 (20.3%)
C4–C5	3	3	18	24 (40.7%)
C5–C6	4	5	19	28 (47.5%)
C6–C7	2	3	7	12 (20.3%)
C7–T1	0	1	2	3 (5.1%)
**Parameter**	**Group A** **(*n* = 9)**	**Group B** **(*n* = 13)**	**Group C** **(*n* = 37)**	** *p* ** **-Value**
**Epidural abscess**	0	3	21	0.001 †
**Location of infection**				
Anterior	9	12	35	0.649 †
Posterior	0	1	1	
Anteroposterior	0	0	1	
**Multisegmental infection**	1	2	17	0.017 †

† Linear by linear association. Location of infection: anterior = anterior epidural space or vertebral body or intervertebral disc, posterior = posterior epidural space or lamina or spinous process or transverse process.

**Table 6 jcm-11-02210-t006:** Complications.

Parameter	Group A (*n* = 9)	Group B (*n* = 13)	Group C (*n* = 37)	*p*-Value
**Total complications**	1/9	2/13	9/37	0.321 †
Reoperation	0	2	3	
Superficial surgical site infection	0	0	1	
Esophageal fistula	0	0	1	
Dysphagia	0	0	1	
Voice change	1	0	0	
Neurological impairment	0	0	1	
Death	0	0	4	0.147 †

† Linear by linear association.

## Data Availability

The data was not publicly available due to ethical reasons and patient privacy.

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
