# Peer review of "Chronological Analysis of Primary Cervical Spine Infection: A Single-Center Analysis of 59 Patients over Three Decades (1992–2018)"

_jcm, 2022, doi:10.3390/jcm11082210_

Round 1

Reviewer 1 Report

The diagnostic methods for patients with suspected cervical spine infection (CSI) between years can influence the incidence and the number of patients with such infections. In other words, it is unlikely that during the study period, all physician used similar diagnostic methods for diagnosing spinal lesions. Furthermore, patients were retrieved by ICD9 numbers; for such identification, patients may be misclassified with or without CSI (No pathogen was isolated from most patients).

The other concern is the number of patients is too small. 

The arrangement of parameters in the Table 1 & 3 is confused and should be corrected. 

In the Table 2, the presentation of "mild fever" is not clear. 

Reviewer 2 Report

The work is very interesting and deals with the subject of primary spine infections in the cervical region. The paper is presented in an interesting, clear and easy-to-read manner.

In the part clinical analysis, there is no description of the type of paresis ; spastic or flaccid paresis, as many muscle groups are affected by paresis , whether there were sensory disturbancesThere are no comments on the clinical symptoms of spine infection in the discussion section. It should be emphasized that the pains in the cervical region, which are typical for an infection of the spine in the cervical region, may also result from the degenerative changes of the spine, which are very common in the elderly. However, pain from infection is of a different nature, it is usually constant, continuous and does not change during the dayIn my opinion, doctors must  be very vigilant during examining patients with these ailments in order to promptly refer patients to MRI in the case of atypical ailments.Although the authors pay attention to the rapid diagnosis and treatment of these infections, but there is no description of clinical symptoms indicating cervical spine infections.
